# HIV-1 Env trimer opens through an asymmetric intermediate in which individual protomers adopt distinct conformations

Xiaochu Ma[1], Maolin Lu[1], Jason Gorman[2], Daniel S Terry[3], Xinyu Hong[1], Zhou Zhou[3], Hong Zhao[3], Roger B Altman[3], James Arthos[4], Scott C Blanchard[3], Peter D Kwong[2], James B Munro[5]*, Walther Mothes[1]*

[1]Department of Microbial Pathogenesis, Yale University School of Medicine, New Haven, United States; [2]Vaccine Research Center, National Institute of Allergy and Infectious Diseases, National Institutes of Health, Bethesda, United States; [3]Department of Physiology and Biophysics, Weill Cornell Medical College of Cornell University, New York, United States; [4]Laboratory of Immunoregulation, National Institute of Allergy and Infectious Diseases, National Institutes of Health, Bethesda, United States; [5]Department of Molecular Biology and Microbiology, Tufts University School of Medicine, Boston, United States

*For correspondence:
james.munro@tufts.edu (JBM);
walther.mothes@yale.edu (WM)

**Abstract** HIV-1 entry into cells requires binding of the viral envelope glycoprotein (Env) to receptor CD4 and coreceptor. Imaging of individual Env molecules on native virions shows Env trimers to be dynamic, spontaneously transitioning between three distinct well-populated conformational states: a pre-triggered Env (State 1), a default intermediate (State 2) and a three-CD4-bound conformation (State 3), which can be stabilized by binding of CD4 and coreceptor-surrogate antibody 17b. Here, using single-molecule Fluorescence Resonance Energy Transfer (smFRET), we show the default intermediate configuration to be asymmetric, with individual protomers adopting distinct conformations. During entry, this asymmetric intermediate forms when a single CD4 molecule engages the trimer. The trimer can then transition to State 3 by binding additional CD4 molecules and coreceptor.
DOI: https://doi.org/10.7554/eLife.34271.001

## Introduction

The human immunodeficiency virus 1 (HIV-1) enters CD4+ T cells through the interaction of its envelope glycoprotein (Env) with cell-surface receptor CD4 and coreceptor CCR5 or CXCR4 (*Wyatt and Sodroski, 1998*; *Doms and Moore, 2000*). Env is a trimer with each protomer consisting of two subunits: the surface subunit gp120 that binds to receptor and coreceptor, and the transmembrane subunit gp41 that mediates fusion of viral and cellular membranes. CD4 binding induces conformational changes in Env that expose structural elements required for coreceptor binding including the V3 loop and the bridging sheet (*Trkola et al., 1996*; *Kwong et al., 1998*; *Huang et al., 2007*; *Liu et al., 2008*; *Pancera et al., 2010*; *Wang et al., 2016*; *Herschhorn et al., 2017*; *Ozorowski et al., 2017*). Binding of coreceptor triggers additional conformational changes in gp41, including the formation of an extended gp41 structure that subsequently collapses into a stable six-helix bundle, which is thought to drive viral and cellular membranes together for fusion (*Pancera et al., 2010*; *Blumenthal et al., 2012*; *Harrison, 2015*). Correspondingly, Env's capacity to undergo extensive conformational changes is critical for virus entry.

At the same time, Env evades immune surveillance through 'conformational masking' (*Kwong et al., 2002*), which protects key functional elements within the trimer from being recognized by antibodies. This feature renders most Env-targeting antibodies non-neutralizing. However, a portion of patients develop potent broadly neutralizing antibodies that can prevent immunodeficiency virus infections in animal models, lower the viral load when administered to HIV-1-infected patients and can restore immunological control in the absence of antiretroviral therapy (ART) in non-human primates (*Wu et al., 2010*; *Walker et al., 2011*; *Klein et al., 2012*; *Caskey et al., 2015*; *Gautam et al., 2016*; *Lu et al., 2016*; *Schoofs et al., 2016*; *Nishimura et al., 2017*). Antibodies that are broadly neutralizing tend to recognize closed Env conformations (*Munro et al., 2014*; *Guttman et al., 2015*). Substantial efforts have been made to characterize structurally closed Env trimers using stabilized soluble ectodomains as well as detergent-solubilized Env proteins (*Julien et al., 2013*; *Lyumkis et al., 2013*; *Pancera et al., 2014*; *Lee et al., 2016*). However, how the Env trimer opens, and through what structural intermediates it transitions, is poorly understood.

We have previously applied single-molecule Fluorescence Resonance Energy Transfer (smFRET) imaging to visualize the dynamics of individual Env molecules on the surface of native virions of two HIV-1 strains, NL4-3 and JR-FL (*Munro et al., 2014*). Donor and acceptor fluorophores were introduced into the variable loops V1, and V4 or V5 of a single gp120 subunit in an otherwise unlabeled virus. smFRET analysis revealed that single Env protomers spontaneously transit between three distinct conformational states exhibiting low-, intermediate- and high-FRET values (*Munro et al., 2014*). The unliganded Env prefers a low-FRET pre-triggered conformation. In the presence of soluble CD4 (D1D2 domain, sCD4), and the additional presence of the coreceptor-surrogate antibody 17b, some Env trimers could be stabilized in high-FRET and intermediate-FRET conformations, respectively (*Munro et al., 2014*).

The FRET-indicated Env conformational states initially observed require further structural assignments. Various lines of evidence suggest that the low-FRET state corresponds to the pre-triggered conformation: (1) it is the most populated conformation of the unliganded Env; (2) it is more populated in a clinical isolate such as HIV-1$_{JR-FL}$ that is more neutralization-resistant than the laboratory-adapted HIV-1$_{NL4-3}$; and (3) it is stabilized by broadly neutralizing antibodies, and the small-molecule conformational blocker BMS-626529 (*Munro et al., 2014*; *Pancera et al., 2014*; *Kwon et al., 2015*; *Herschhorn et al., 2017*; *Pancera et al., 2017*). In contrast, the structural nature of the intermediate- and high-FRET states has been unclear. The stabilization of the high-FRET State two in HIV-1$_{NL4-3}$ by sCD4 and of intermediate-FRET State three by sCD4/17b suggested that they might represent CD4 or coreceptor-bound conformations, respectively. However, this conflicts with the finding that the CD4 mimetic JRC-II-191 stabilizes intermediate-FRET configurations (State 3) (*Munro et al., 2014*), as we would expect CD4 mimetics to reproduce the conformational impact of CD4. Also, there are no substantial Env-structural differences between the gp120 bound to CD4 or to CD4 and 17b (*Kwong et al., 1998*; *Ozorowski et al., 2017*). Finally, HIV-1$_{JR-FL}$ does not respond to sCD4 in the same way as HIV-1$_{NL4-3}$.

Here we provide smFRET analysis that clarify the nature of the intermediate-FRET and high-FRET Env conformations. Through smFRET measurements, we demonstrated that dodecameric CD4 (sCD4$_{D1D2}$-Igαtp), which is a potent neutralizer of HIV-1, stabilizes the intermediate-FRET State 3 of the Env in three strains HIV-1$_{NL4-3}$, HIV-1$_{JR-FL}$ and HIV-1$_{BG505}$. We observed no difference in predominant conformational state of Env bound to CD4/17b, to JRC-II-191 (a CD4 small molecule mimetic), or to sCD4$_{D1D2}$-Igαtp, all of which corresponded to the open three-CD4-bound conformation of the trimer (State 3). Moreover, by using a mixed trimer assay, we identified the intermediate configuration of State two to be an asymmetric Env trimer. In this configuration, a single CD4 can engage the trimer such that individual protomers adopt distinct conformations. An asymmetric trimer is thus a prevalent functional intermediate, which can open further either spontaneously or through binding to additional CD4 molecules or to coreceptor.

## Results

### State 3 corresponds to the gp120 conformation of the three-CD4-bound HIV-1 Env trimer

smFRET imaging of HIV-1 Env trimers carrying a single pair of donor and acceptor fluorophores in the V1 and V4 loops of gp120, previously established for HIV-1 isolates NL4-3 and JR-FL (*Munro et al., 2014*), was extended to include the mother-to-child transmitted founder (T/F) virus BG505 widely used for structural studies (*Figure 1—figure supplement 1*) (*Julien et al., 2013*; *Lyumkis et al., 2013*; *Sanders et al., 2013*; *Pancera et al., 2014*; *Kwon et al., 2015*; *Scharf et al., 2015*; *Wang et al., 2016*; *Ozorowski et al., 2017*). The insertion of Q3 and A1 tags into the variable loops V1 and V4 of gp120 for enzymatic labeling was validated by infectivity assays, protein expression and antibody neutralizations, indicating that the Env functions remained minimally affected even in the 100% dually tagged virus (*Figure 1—figure supplements 2* and *3*).

Surface-bound HIV-1$_{NL4-3}$, HIV-1$_{JR-FL}$ and HIV-1$_{BG505}$ viruses carrying a single dually-tagged Env were imaged by total internal reflection fluorescence (TIRF) microscopy. In agreement with previous results (*Munro et al., 2014*), FRET trajectories and histograms showed that the gp120 protomers in unliganded HIV-1 Env predominantly resided in a low-FRET conformation, but had inherent access to both intermediate- and high-FRET conformations (*Figure 1A and B*, left). Consistent with the features of Tier two viruses, the HIV-1 isolates JR-FL and BG505 (*Koyanagi et al., 1987*; *Wu et al., 2006*) exhibited a higher occupancy of the pre-triggered low-FRET state as compared to the lab-adapted NL4-3 (compare *Figures 1E and H, B*).

We applied Hidden Markov Modeling (HMM) using a three-state model to analyze the sequence of transitions between the conformational states and displayed them in a transition density plot (TDP) (*Figure 1B*, right). The unliganded HIV-1$_{NL4-3}$ Env frequently transitioned between the low-FRET and high-FRET states, and between the high- and intermediate-FRET states; whereas transitions between low- and intermediate-FRET were rarely observed. Given that the low-FRET state corresponds to a mature pre-triggered conformation of the trimer, these data suggest that the high-FRET state represents a default conformational intermediate during opening of the Env trimer. Based on this sequence of FRET transitions, we refer to the low-FRET state as State 1, the high-FRET state as State two and the intermediate-FRET state as State 3 (*Herschhorn et al., 2016*).

Consistent with previous observations, addition of sCD4 (0.1 mg/ml) stabilized HIV-1$_{NL4-3}$ Env in State 2 (*Figure 1C*, left). The predominant FRET transitions observed were between States 2 and 3. The observed stabilization of State two by sCD4 was initially difficult to reconcile with the observed State three stabilization by the CD4-mimetic small molecule, JRC-II-191, as well as the combination of sCD4 and the coreceptor-surrogate antibody 17b (*Munro et al., 2014*). To clarify the response of HIV-1$_{NL4-3}$ Env to receptor CD4, a dodecameric CD4 oligomer (sCD4$_{D1D2}$-Igαtp, 0.1 mg/ml), which is 300 to 1000-fold more potent in neutralizing HIV-1 than sCD4, was used to provide a high local CD4 density (*Arthos et al., 2002*; *Kwong et al., 2002*). Strikingly, sCD4$_{D1D2}$-Igαtp stabilized HIV-1$_{NL4-3}$ Env in State 3 (*Figure 1D*), indicating that both sCD4/17b (*Munro et al., 2014*) and an oligomerized CD4, both stabilize HIV-1$_{NL4-3}$ Env in the three-CD4-bound trimer configuration (State 3).

In contrast to the NL4-3 isolate, both HIV-1$_{JR-FL}$ and HIV-1$_{BG505}$ were less responsive to sCD4, with only small increases in the occupancies of States 2 and 3 (*Figure 1F and I*) although sCD4 and 17b stabilized both in State 3 (*Munro et al., 2014*) (*Figure 1—figure supplement 4*). Importantly, as was observed for HIV-1$_{NL4-3}$ Env, addition of the dodecameric sCD4$_{D1D2}$-Igαtp stabilized both HIV-1$_{JR-FL}$ and HIV-1$_{BG505}$ in State 3 (*Figure 1G and J*). This suggests that for all three HIV-1 isolates the intermediate-FRET State 3 corresponds to the three-CD4-bound conformation (*Figure 1K*).

### State 2 arises from an asymmetric trimer, in which a single CD4 molecule engages HIV-1 Env

The assignment of the three-CD4-bound conformation as State 3 leaves the nature of the structural intermediate State 2 unresolved. We used the stabilization of HIV-1$_{NL4-3}$ State 2 by sCD4 as an assay to identify the origin of the high-FRET signal. We hypothesized that the different conformational effects of sCD4 and sCD4$_{D1D2}$-Igαtp on HIV-1$_{NL4-3}$ Env are linked to the differences in CD4 occupancy. The dodecameric sCD4$_{D1D2}$-Igαtp, with its higher avidity and local CD4 density, may engage two or three protomers of the trimer (*Arthos et al., 2002*; *Bennett et al., 2007*) while monomeric

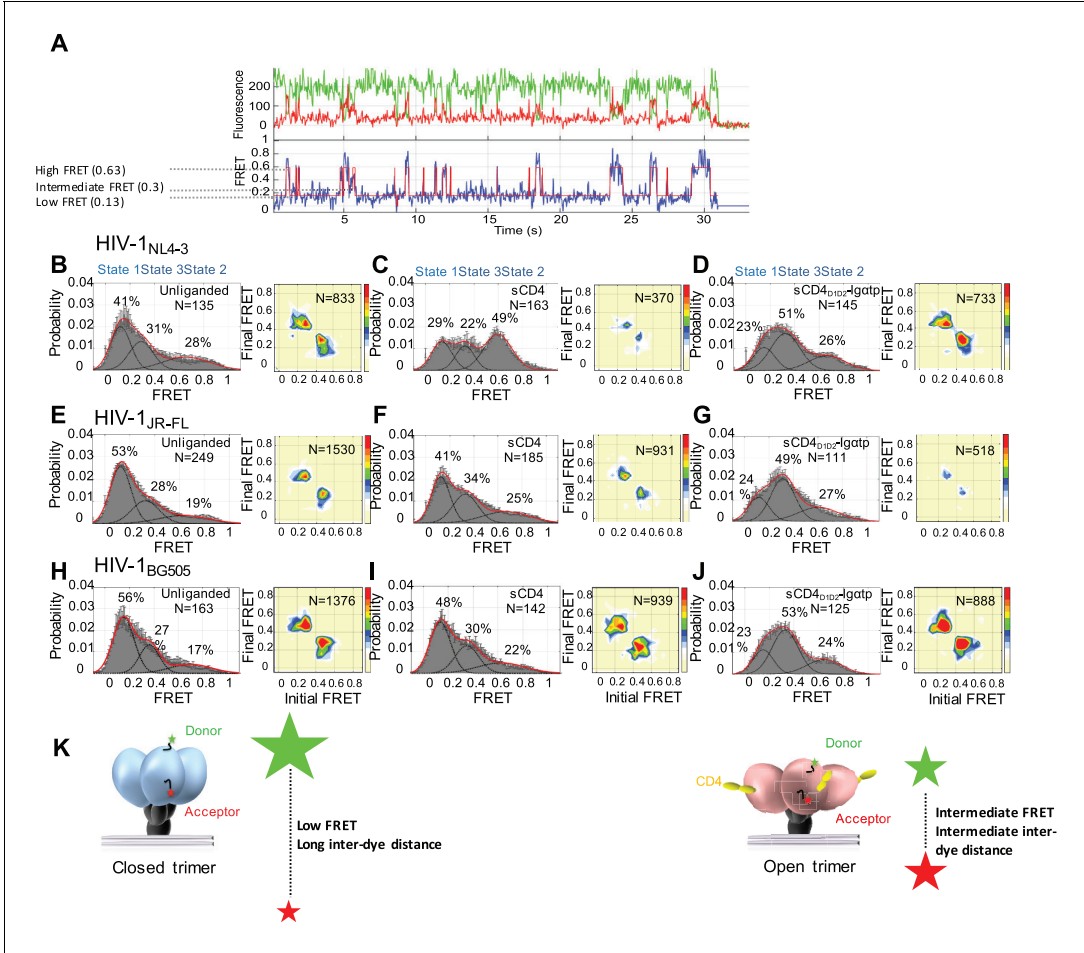

**Figure 1.** State 3 corresponds to the gp120 conformation of the three-CD4-bound HIV-1 Env trimer. (**A**) Representative smFRET trace for unliganded HIV-1$_{NL4-3}$ Env. (Top) The donor fluorophore (green) was attached to the V1 loop and the acceptor fluorophore (red) was attached to the V4 loop. (Bottom) Corresponding FRET trajectory (blue) with overlaid idealization generated by Hidden Markov Modeling (HMM) (red). (**B**) (Left) Probability distribution of FRET values compiled from all the individual HIV-1$_{NL4-3}$ Env molecules (N = number of FRET traces analyzed). The histogram was fitted to a sum of three Gaussian distributions, with means of 0.13, 0.3 and 0.63, which are corresponding to States 1, 3 and 2, as indicated. The percentage indicates the occupancy of each FRET state. Error bars represent standard errors calculated from histograms from three independent sets of FRET traces. (Right) Transition Density Plot (TDP) of all the observed transitions in unliganded HIV-1$_{NL4-3}$ Env. Color bar shows the scale used to indicate the frequency of each transition. (**C, D**) sCD4 (0.1 mg/ml) (**C**) or sCD4$_{D1D2}$-Igαtp (0.1 mg/ml) (**D**) was incubated with the virus for 30 min prior to imaging. FRET histogram and TDP are as in (**B**). (**E–G**) Probability distributions (left) and TDPs (right) for HIV-1$_{JR-FL}$ Env for the unliganded (**E**) (Note: FRET histogram and TDP were from previous data set for direct comparison [*Herschhorn et al., 2016*]), sCD4-bound (0.1 mg/ml) (**F**) and sCD4$_{D1D2}$-Igαtp-bound (0.1 mg/ml) (**G**). (**H–J**) Probability distributions (left) and TDPs (right) for HIV-1$_{BG505}$ Env for the unliganded (**H**), sCD4-bound (0.1 mg/ml) (**I**) and sCD4$_{D1D2}$-Igαtp-bound (0.1 mg/ml) (**J**) viruses are displayed as in HIV-1$_{NL4-3}$. (**K**) Schematic illustration of the closed and open conformations of the Env trimer. The unliganded conformation is in blue and CD4-bound conformation is in pink. Green and red starts represent donor and acceptor fluorophores, respectively. Sizes of the stars represent relative change of fluorescence between donor and acceptor dyes and dotted line indicated changes of inter-dye distances.

DOI: https://doi.org/10.7554/eLife.34271.002

The following figure supplements are available for figure 1:

**Figure supplement 1.** Peptide insertion sites into the V1 and V4 loops of gp120 of three HIV-1 isolates.

DOI: https://doi.org/10.7554/eLife.34271.003

**Figure supplement 2.** Infectivity and Env incorporation of single or dually tagged HIV-1$_{BG505}$ viruses.

DOI: https://doi.org/10.7554/eLife.34271.004

**Figure supplement 3.** Dually tagged HIV-1$_{BG505}$ antibodies are neutralized by trimer specific antibodies.

DOI: https://doi.org/10.7554/eLife.34271.005

**Figure supplement 4.** smFRET histogram for HIV-1$_{BG505}$ Env bound to sCD4 and 17b.

DOI: https://doi.org/10.7554/eLife.34271.006

sCD4 binds with less avidity and may only interact with one protomer. The distinct high-FRET conformation of State 2 may then arise from either a distinct conformation in the first CD4-bound protomer, or from neighboring gp120 subunits.

To distinguish between these possibilities, we took advantage of the D368R mutation in the CD4-binding pocket of Env that reduces affinity for CD4 (*Olshevsky et al., 1990*). The D368R mutant of HIV-1$_{NL4-3}$ Env was resistant to the neutralization of both sCD4 and sCD4$_{D1D2}$-Igαtp (*Figure 2A*, Table 2). The mutant Env was expressed similarly to that of WT Env (*Figure 2—figure supplement 1*). Their infectivity was reduced by ~100 fold, but the signal to noise ratio in our experiments was orders of magnitude above background permitting a neutralization assay and demonstrating the resistance of D368R to sCD4 (*Figure 2—figure supplement 2*). Trimers that uniformly contained the D368R mutation exhibited similar conformational landscapes to that of wild-type HIV-1, and did not respond to sCD4 and sCD4$_{D1D2}$-Igαtp as WT HIV-1$_{NL4-3}$ (*Figure 2—figure supplement 3A*). Thus, unlike the double mutant D368R/E370R that was stabilized in State 1 (*Munro et al., 2014*) (likely due to E370R interfering with the adoption of the bridging sheet), the single D368R mutation exhibited little or no conformational effects on the trimer. This allowed us to engineer a 'mixed trimer 1' where the two unlabeled protomers carried the D368R mutation to prevent binding of CD4 and the single protomer carrying the fluorophores remained competent for CD4 binding (*Figure 2B*). smFRET analysis for the mixed HIV-1$_{NL4-3}$ trimer 1 revealed a clear State 3 stabilization by sCD4 (*Figure 2D*). The occupancy of State 2 was similar to the native HIV-1$_{NL4-3}$ trimer bound by sCD4$_{D1D2}$-Igαtp (for comparison, see *Figure 1D*). Thus, gp120 bound to CD4 adopts a State 3 conformation regardless of the CD4 occupancy elsewhere within the trimer, even when only a single CD4 molecule engages the trimer.

We next examined the conformation of the CD4-binding incompetent protomers (carrying a D368R mutation) next to the single-CD4-bound protomer, which we designated 'mixed trimer 2'. To this end, HEK293 cells were co-transfected with a 1:1 ratio of plasmid encoding HIV-1 D368R mutant and wild-type HIV-1$_{NL4-3}$ Env in excess over plasmid encoding HIV-1$_{D368R}$ with dually tagged Env. As a result,~50% of the trimers were expected to carry fluorophores in the CD4-binding incompetent D368R protomer adjacent to a single protomer that can bind CD4 (*Figure 2C*). Under these conditions,~25% of trimers were expected to carry the D368R mutation in all three protomers. This subpopulation cannot bind CD4. The remaining 25% carry two CD4-binding competent protomers next to the labeled mutant gp120 (*Figure 2C*).

Strikingly, despite the heterogeneity in the population of trimers, imaging of the HIV-1$_{NL4-3}$ mixed trimer two restored the high occupancy of the State two conformation (*Figure 2E*). Hence, the observed high-FRET state likely originates from an unbound gp120 protomer adjacent to a CD4-bound protomer. Since the State three and State two stabilizations were observed for the same single-CD4 condition, but with dyes placed either into the protomer that binds a single CD4 or adjacent to it, this trimer must be asymmetric with protomers adopting distinct conformations (*Figure 2L*).

We next similarly generated mixed Env trimers for the Tier 2 HIV-1 isolates HIV-1$_{JR-FL}$ and HIV-1$_{BG505}$. Since these Tier two viruses were less responsive to sCD4, we used the more potent ligand dodecameric sCD4$_{D1D2}$-Igαtp. The D368R mutants for both HIV-1 isolates made all Envs largely resistant to sCD4 and sCD4$_{D1D2}$-Igαtp (*Figure 2F and I* and *Figure 2—figure supplement 3B and C*). Intriguingly, the engineering of mixed trimers that can only bind a single CD4 allowed the asymmetric trimer to be stabilized, confirming that the high-FRET of the default structural intermediate originates from protomers adjacent to a single bound CD4 molecule (*Figure 2G,H,J,K*). The broad high-FRET peaks in sCD4$_{D1D2}$-Igαtp-bound mixed trimer two suggest a potential continuum of various conformational states. These data support the hypothesis that the trimer spontaneously opens and closes through an asymmetric trimer configuration. Moreover, during virus entry, a single CD4 may initially engage a closed trimer (*Kwon et al., 2015*; *Hu et al., 2017*), and a very early version of this intermediate may be presented in the single CD4-bound structure of the Env trimer stabilized by the DS-SOSIP (*Liu et al., 2017*) when not fixed by DS mutation, CD4 binding to the trimer would induce the CD4-bound conformation in the bound protomer (*Figure 2L*). Binding of a single CD4 may loosen interaction of the V1/V2 loops in the trimer association domain so that neighboring protomers can adopt a conformation in which the V1 and V4 loops are closer to each other.

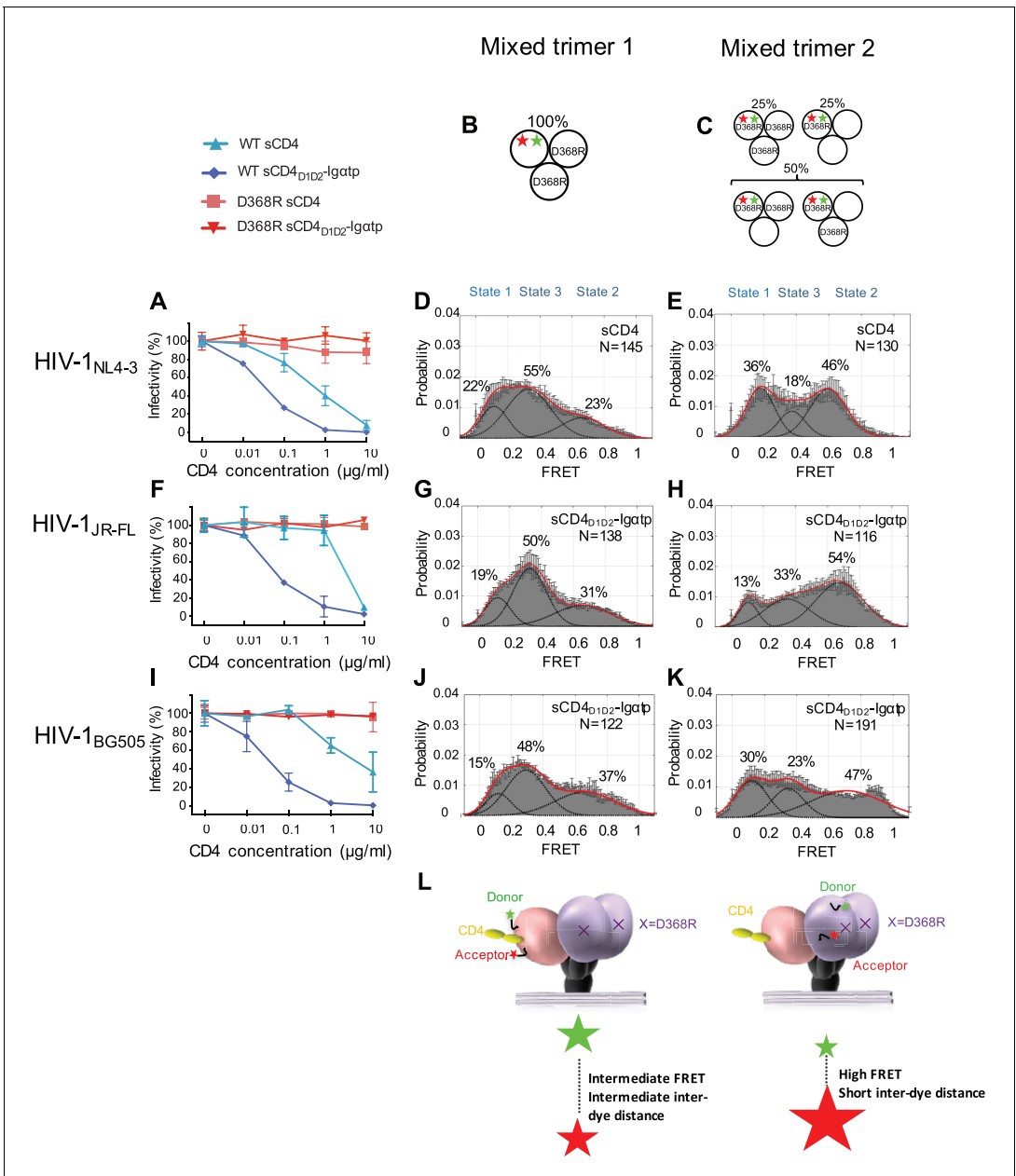

**Figure 2.** State 2 corresponds to an asymmetric trimer, in which a single CD4 molecule engages HIV-1 Env. (**A**) Neutralization curves of WT and D368R HIV$_{NL4-3}$ viruses by sCD4 and sCD4$_{D1D2}$-Igαtp. Data represent three independent experiments ± standard deviation. (**B**) Scheme to illustrate generation of mixed HIV-1 Env trimer 1, in which the two unlabeled protomers contained the D368R mutation to prevent CD4 binding, and the CD4-binding competent WT protomer carried the donor and acceptor fluorophores (green, red stars in scheme above). (**C**) Scheme to illustrate generation of mixed HIV-1 trimer 2, in which sCD4 can only engage gp120 domains adjacent to the labeled domain. Given the co-transfection protocol of indicated HIV-1 plasmids, only 50% of all trimers are expected to exhibit this configuration. 25% of trimers are expected to carry D368R mutation in all three protomers and the remaining 25% would carry two CD4-binding competent protomers next to the labeled mutant gp120. (**D**) FRET histogram as in *Figure 1* for the mixed HIV-1$_{NL4-3}$ Env trimer 1. (**E**) FRET histogram for the mixed HIV-1$_{NL4-3}$ Env trimer 2. Neutralization curves for HIV-1$_{JR-FL}$ (**F**) or HIV-1$_{BG505}$ (**I**) were shown as in HIV-1$_{NL4-3}$. FRET histograms for mixed HIV-1$_{JR-FL}$ (**G–H**) and HIV-1$_{BG505}$ (**J–K**) Env trimer 1 and 2 are shown as HIV-1$_{NL4-3}$. Viruses were incubated with sCD4 (0.1 mg/ml) or sCD4$_{D1D2}$-Igαtp (0.01 mg/ml) for 30 min prior to imaging as indicated. (**L**) Schematic illustration of the asymmetric opening of the Env trimer. The CD4-bound conformation is in pink, and the conformational intermediate in the asymmetric trimer is in purple. The purple x indicates the D368R mutation. Green and red stars represent donor and acceptor fluorophores, respectively. Sizes of the stars represent relative change of fluorescence between donor and acceptor dyes and dotted line indicated changes of inter-dye distances.

DOI: https://doi.org/10.7554/eLife.34271.007

The following figure supplements are available for figure 2:

*Figure 2 continued on next page*

*Figure 2 continued*

**Figure supplement 1.** D368R carrying Envs are expressed and incorporated into virions similar to wild-type.
DOI: https://doi.org/10.7554/eLife.34271.008
**Figure supplement 2.** Infectivity of HIV-1$_{D368R}$ viruses.
DOI: https://doi.org/10.7554/eLife.34271.009
**Figure supplement 3.** D368R inhibits sCD4 and sCD4$_{D1D2}$-Igαtp binding of all three HIV-1 Envs.
DOI: https://doi.org/10.7554/eLife.34271.010

## Binding of additional CD4 molecules or coreceptor surrogate antibody 17b completely opens the Env trimer

To further investigate how the conformation of Env changes when an additional CD4 binds to the trimer, we designed 'mixed trimer 3' for all three isolates, where two protomers are competent for binding to CD4, but the donor and acceptor fluorophores reside in the single protomer carrying a D368R mutation (*Figure 3A*). This allowed us to detect the conformation of the ligand free protomer when two CD4 molecules bind to the other two protomers within the Env trimer. Interestingly, when two CD4 molecules are bound to the Env trimer, the ligand-free protomer adopted the State three conformation, indicating that the trimer had fully opened (*Figure 3B–D*). Binding of two CD4 molecules is enough to flip the third protomer open indicating cooperativity between the three protomers. Our data are consistent with previous reports on the necessity of binding of multiple CD4 molecules to Env, as well as CD4 clustering (*Yang et al., 2006*; *Salzwedel and Berger, 2009*).

We then asked if the asymmetric trimer intermediate, containing a single CD4-bound protomer (State two predominant configuration), is competent for coreceptor binding. To this end, we imaged the mixed trimer 2 of all three HIV-1 isolates and incubated them with both sCD4 (0.1 mg/ml) and coreceptor surrogate antibody 17b (0.1 mg/ml) (*Figure 3E*). We observed that the mixed trimer two bound by both a single sCD4 molecule and 17b also adopted the State three conformation (*Figure 3F–H*). This suggests that coreceptor binding would also be sufficient to trigger the complete opening of the single-CD4 bound asymmetric trimer (*Figure 3I*, *Figure 3—figure supplement 1*).

## Kinetic and thermodynamic analysis of smFRET data

We performed a full kinetic and thermodynamic analysis of the FRET trajectories to quantify how ligand binding remodels the energy landscape governing Env dynamics. This analysis was permitted by organic fluorophores with enhanced photostability (*Zheng et al., 2014*), as well as the application of sCMOS cameras, which offered access to greater numbers of FRET trajectories (*Juette et al., 2016*). The occupancies in the FRET states were determined from the FRET histograms, and used to calculate the differences in free energies between states $i$ and $j$ according to $\Delta G°_{ij} = -k_B T \ln(P_i/P_j)$, where $P_i$ and $P_j$ are the occupancies of the $i$th and $j$th state in the histogram, respectively, and $k_B$ is the Boltzmann constant.

In HIV-1$_{NL4-3}$, the predominant effect of sCD4 binding was to profoundly stabilize State 2. Before binding of sCD4, the relative free energy of HIV-1$_{NL4-3}$ State 2 was 0.37 $k_B T$ higher than State 1 (*Figure 4A*, left). Following binding of sCD4, the free energy of State 2 was 0.54 $k_B T$ lower than State 1, resulting in a net change of −0.91 $k_B T$. With respect to activation energies, sCD4 binding decreased the activation energy for the transitions from State one and State three into State 2, and increased the activation energy for transitions out of State 2. Along with the thermodynamic stabilization of State 2, these data offer an energetic explanation for the accumulation of molecules in State 2 (*Figure 1C*). In contrast, the predominant effect of sCD4$_{D1D2}$-Igαtp binding was to stabilize State 3 (*Figure 4A*, right, *Figure 1D*). Before sCD4$_{D1D2}$-Igαtp binding, the free energy of State 3 was 0.11 $k_B T$ lower than State 2. Following sCD4$_{D1D2}$-Igαtp binding the free energy of State 3 was 0.69 $k_B T$ lower than the liganded State 2. With the liganded State 2 being 0.28 $k_B T$ lower than unliganded, this results in a net change of −0.86 $k_B T$.

Next, the dwell times in each FRET state identified by the application of HMM were compiled into histograms and fit to exponential distributions (*Figure 4—figure supplement 1*), revealing the rates of transition between the FRET states (*Table 1*). The rate constants obtained from this analysis were used to calculate the change in activation energies between two states $\Delta \Delta G^{\ddagger}_{ij} = -k_B T$ ln

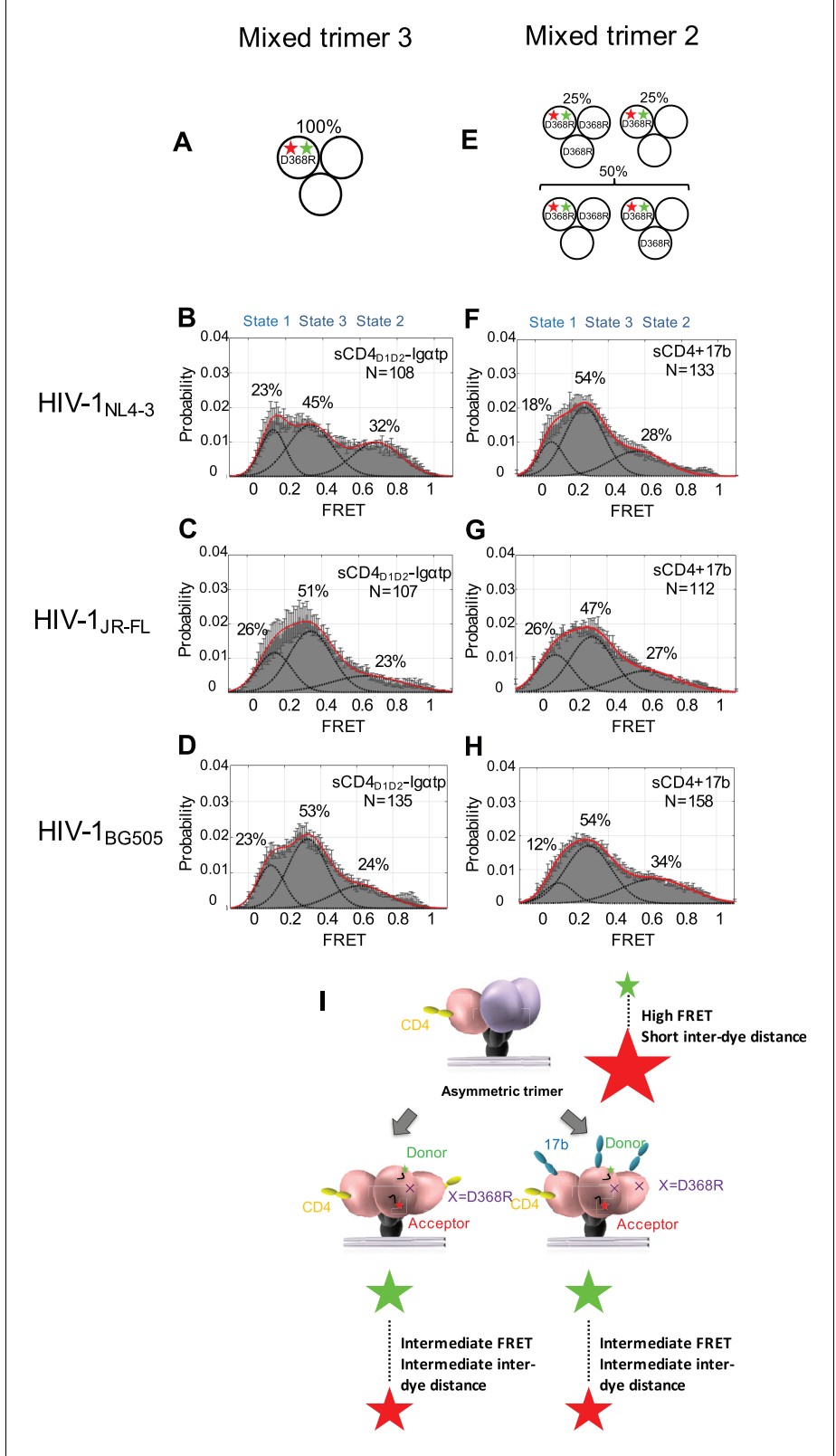

**Figure 3.** Binding of additional CD4 molecules or coreceptors completely opens the Env trimer. (**A**) Scheme to illustrate generation of mixed HIV-1 Env trimer 3, in which two unlabeled protomers are CD4-binding competent and the one protomer carrying the donor and acceptor fluorophores (green, red stars in scheme above) is CD4-binding defective because it has the D368R mutation. (**B–D**) FRET histograms as in **Figure 1** for the mixed HIV-1

*Figure 3 continued on next page*

*Figure 3 continued*

Env trimer 3. sCD4$_{D1D2}$-Igαtp (0.01 mg/ml) was incubated with the virus for 30 min prior to imaging. (E) Scheme to illustrate generation of mixed HIV-1 Env trimer 2, as in *Figure 2*. (F–H) FRET histograms as in *Figure 1* for the mixed HIV-1 Env trimer 2. sCD4 (0.1 mg/ml) and 17b (0.1 mg/ml) were incubated with the virus for 30 min prior to imaging. (I) Schematic illustration of the further activation of the Env trimer. from the asymmetric intermediate. The CD4-bound conformation is in pink, and the conformational intermediate in the asymmetric trimer is in purple. The purple x indicates the D368R mutation. Green and red starts represent donor and acceptor fluorophores, respectively. Sizes of the stars represent relative change of fluorescence between donor and acceptor dyes and dotted line indicated changes of inter-dye distances.

DOI: https://doi.org/10.7554/eLife.34271.011

The following figure supplement is available for figure 3:

**Figure supplement 1.** Model for the activation of the HIV-1 Env trimer through asymmetric intermediates.

DOI: https://doi.org/10.7554/eLife.34271.012

---

($k_{ij}^{liganded}/k_{ij}^{unliganded}$), where $k_{ij}$ is the rate of transition from the $i$th to $j$th FRET state. According to this analysis, sCD4 binding decreased the activation energy for the transitions from State one and State three into State 2, and increased the activation energy for transitions out of the State 2. Along with the thermodynamic stabilization of State 2, these data offer an energetic explanation for the accumulation of molecules in State 2. In contrast, the major effect of sCD4$_{D1D2}$-Igαtp binding was the thermodynamic stabilization of State 3, with only minor effects on the observed kinetics (*Figure 4A*, right).

We then repeated these experiments for the clinical HIV-1 isolate JR-FL and BG505. The unliganded HIV-1$_{JR-FL}$ and HIV-1$_{BG505}$ Envs more stably adopted State one as compared to the HIV-1$_{NL4-3}$ Env. This finding indicates that, relative to the other states, the closed state has lower energy in HIV-1$_{JR-FL}$ compared with the lab-adapted HIV-1$_{NL4-3}$ isolate (*Figure 4B and C*, *Figure 1E and H*). Specifically, the energy of State 1 of HIV-1$_{JR-FL}$ and HIV-1$_{BG505}$ was lower than that of State 2 by 1.05 $k_BT$ and 1.16 $k_BT$, respectively, as compared to 0.37 $k_BT$ for NL4-3 (*Figure 4B and C*). We also found a modest decrease in the activation energy for the sCD4-induced transition out of State 1, and decreased activation energy for exchanges between States 2 and 3 (*Figure 4B and C*, left). Importantly, as was observed for HIV-1$_{NL4-3}$ Env, addition of the dodecameric sCD4$_{D1D2}$-Igαtp stabilized State 3 (*Figure 4B and C*, right), decreasing the energy to 0.6 $k_BT$ and 0.79 $k_BT$ lower than that of State 2 of HIV-1$_{JR-FL}$ and HIV-1$_{BG505}$, respectively. (*Figure 4B and C*, right).

Interestingly, HIV-1$_{JR-FL}$ and HIV$_{BG505}$ Envs were not as dramatically stabilized in State two by sCD4 binding as was observed for HIV-1$_{NL4-3}$ Env. Rather, binding of sCD4 and sCD4$_{D1D2}$-Igαtp to HIV-1$_{JR-FL}$ and HIV-1$_{BG505}$ Env exhibited similar tendencies in the stabilization of State two as well as State 3. sCD4$_{D1D2}$-Igαtp was clearly more efficient in promoting the accumulation of molecules in the State three conformation. Thus, while there are distinct differences in the ability of sCD4 to activate Env for all three strains, the potent dodecameric sCD4$_{D1D2}$-Igαtp stabilizes Envs from all three isolates in the open conformation (State 3).

## Discussion

Here we used smFRET to identify the nature of the three conformational states sampled by HIV-1 Env (*Munro et al., 2014*). The most populated FRET State one corresponds to the gp120 conformation of the mature pre-triggered Env trimer. The State three stabilization by oligomerized sCD4$_{D1D2}$-Igαtp, sCD4 and 17b, and the CD4 mimetic small molecule JRC-II-191 (*Munro et al., 2014*), indicated that State three corresponds to the gp120 conformation of the open, three-CD4-bound trimer. By using a mixed trimer assay, we identified the most prevalent default intermediate State two to be an asymmetric Env trimer configuration engaging a single CD4 molecule. The individual protomers within the asymmetric trimer adopted different conformations, and State two population originated from the two unbound protomers adjacent to the single CD4-bound protomer.

Based on these single-molecule analyses and previous results, we propose a model of HIV-1 Env activation by receptor CD4 and coreceptor (*Figure 3—figure supplement 1*). This description of the molecular events during Env activation requires a nomenclature that specifies the conformational state of each protomer within the trimer. The unliganded trimer resides predominantly in a

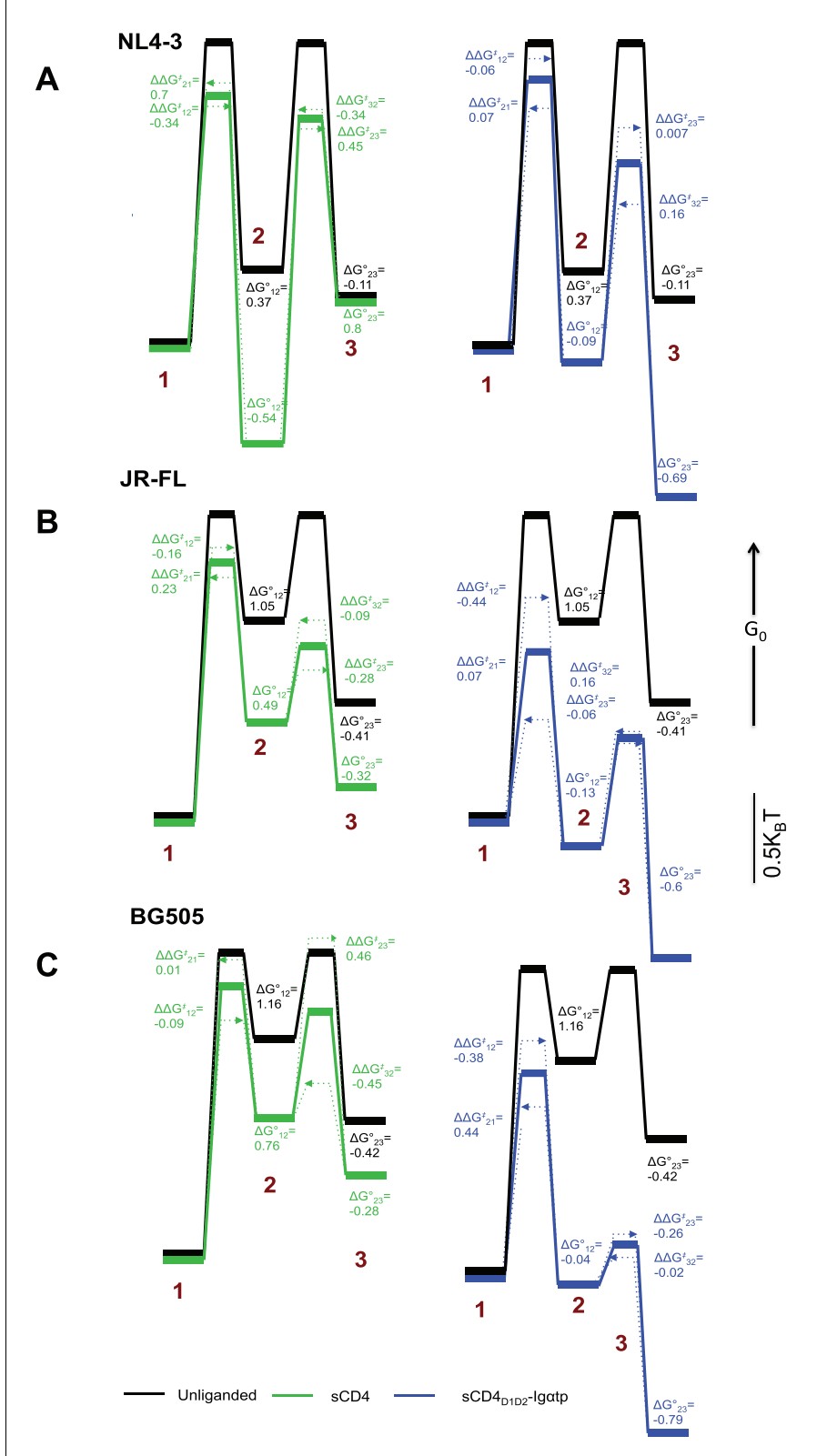

**Figure 4.** Kinetic analysis of HIV-1 Env binding to CD4. Differences in free energies ($\Delta G°_{ij}$) between the FRET states, and the changes in activation energies ($\Delta\Delta G^{\ddagger}_{ij}$) arising from sCD4 and sCD4$_{D1D2}$-Igαtp binding to (**A**) HIV-1$_{NL4-3}$ (**B**) HIV-1$_{JR-FL}$ and (**C**) HIV-1$_{BG505}$. $\Delta G°_{ij}$ values were calculated from the occupancies of each FRET state in the FRET histograms. $\Delta\Delta G^{\ddagger}_{ij}$ values were calculated from the rate constants for the observed transitions for the

*Figure 4 continued on next page*

*Figure 4 continued*

unliganded (black), sCD4-bound (green) and sCD4$_{D1D2}$-Igαtp-bound (blue) HIV-1 Env. The dotted arrow lines indicate the actual $\Delta\Delta G^{\ddagger}_{ij}$ of the forward and backward transitions while the solid line in between indicate the averaged value of the two. Energies are displayed in units of $k_B$T, where $k_B$ is Boltzmann constant, and T is temperature.

DOI: https://doi.org/10.7554/eLife.34271.013

The following figure supplements are available for figure 4:

**Figure supplement 1.** Survival probability distributions for transitions observed in HIV-1$_{NL4-3}$ Env.
DOI: https://doi.org/10.7554/eLife.34271.014
**Figure supplement 2.** Survival probability distributions for transitions observed in HIV-1$_{JR-FL}$ Env.
DOI: https://doi.org/10.7554/eLife.34271.015
**Figure supplement 3.** Survival probability distributions for transitions observed in HIV-1$_{BG505}$ Env.
DOI: https://doi.org/10.7554/eLife.34271.016

conformation in which all three protomers are in the pre-triggered State one configuration, which can be represented as the (1,1,1) configuration (where each of the three numbers defines the FRET state of each protomer within the trimer). Binding of a single CD4 molecule then induces conformational changes in the CD4-bound protomer, such that the trimer adopts a (3,2,2) configuration. In the (3,2,2) configuration, the CD4-bound protomer exhibits intermediate-FRET (State 3), while the two neighboring protomers exhibit high-FRET (State 2). The near absence of direct transitions between State one to State three suggests that the trimer opening likely occurs through a conformational intermediate with a FRET value similar to State two such as a (2,1,1) or (2,2,2) configuration.

The complete activation of HIV-1 Env then likely requires either a local clustering of CD4 or coreceptor binding. This would induce of the Env trimer to reach more open configurations (either (3,3,2) or (3,3,3)) (*Yang et al., 2006*; *Salzwedel and Berger, 2009*; *Ozorowski et al., 2017*). Coreceptor binding then leads to the activation of gp41 and the induction of additional conformational changes necessary for membrane fusion (*Figure 3—figure supplement 1*). During the spontaneous opening and closing of the HIV-1 Env trimer between States 1 and 3, it is possible that the opening protomer does not fully adopt the three-CD4-bound conformation. Rather it may only partially open, just far enough for the trimer apex to be disrupted and the adjacent protomers adopting the State two high-FRET conformation.

**Table 1.** Rates of transition between all observed FRET states for all three HIV-1 Envs.
The distribution of dwell times in each FRET state, determined through Hidden Markov Modeling (HMM), were fit to the sum of two exponential distributions ($y = A_1 \exp^{-k1t} + A_2 \exp^{-k2t}$) (*Figure 4—figure supplements 1–3*). The weighted average of the two rate constants from each fit are presented. Error bars represent 95% confidence intervals propagated from the kinetics analysis.

| HIV-1$_{NL4-3}$ | $k_{1\rightarrow2}$ (s$^{-1}$) | $k_{2\rightarrow1}$ (s$^{-1}$) | $k_{2\rightarrow3}$ (s$^{-1}$) | $k_{3\rightarrow2}$ (s$^{-1}$) |
|---|---|---|---|---|
| Unliganded | 1.50 ± 0.04 | 3.05 ± 0.04 | 1.89 ± 0.05 | 1.32 ± 0.04 |
| sCD4 | 2.10 ± 0.04 | 1.52 ± 0.04 | 1.21 ± 0.03 | 1.85 ± 0.03 |
| sCD4$_{D1D2}$-Igαtp | 1.60 ± 0.02 | 2.84 ± 0.04 | 1.87 ± 0.04 | 1.13 ± 0.02 |
| **HIV-1$_{JR-FL}$** | | | | |
| Unliganded | 1.30 ± 0.03 | 2.21 ± 0.03 | 1.70 ± 0.02 | 1.25 ± 0.02 |
| sCD4 | 1.52 ± 0.03 | 1.76 ± 0.04 | 2.25 ± 0.03 | 1.37 ± 0.05 |
| sCD4$_{D1D2}$-Igαtp | 2.02 ± 0.06 | 2.06 ± 0.05 | 1.81 ± 0.06 | 1.06 ± 0.03 |
| **HIV-1$_{BG505}$** | | | | |
| Unliganded | 1.10 ± 0.04 | 4.37 ± 0.08 | 2.86 ± 0.05 | 1.77 ± 0.03 |
| sCD4 | 1.21 ± 0.03 | 4.32 ± 0.08 | 1.81 ± 0.09 | 2.79 ± 0.04 |
| sCD4$_{D1D2}$-Igαtp | 1.62 ± 0.06 | 2.82 ± 0.05 | 2.92 ± 0.04 | 2.31 ± 0.05 |

DOI: https://doi.org/10.7554/eLife.34271.017

**Table 2.** $IC_{50}$ of sCD4 or sCD4$_{D1D2}$-Ig$\alpha$tp of all three HIV-1 isolates.
Neutralization data were analyzed by nonlinear regression analysis and the Ab concentrations (µg/ml) at which 50% of virus infectivity was inhibited, were calculated.

| | sCD4 | | sCD4$_{D1D2}$-Ig$\alpha$tp | |
|---|---|---|---|---|
| | WT | D368R | WT | D368R |
| HIV-1$_{NL4-3}$ | 0.56 | >50 | 0.03 | >50 |
| HIV-1$_{JR-FL}$ | 3.43 | >50 | 0.06 | >50 |
| HIV-1$_{BG505}$ | 1.26 | >50 | 0.03 | >50 |

DOI: https://doi.org/10.7554/eLife.34271.018

In this paper, we used smFRET to reveal asymmetric trimer configurations in the opening of the HIV-1 Env trimer. Asymmetric structural intermediates during the activation of viral fusion machines may be more common than previously thought. Single receptor engagement and sequential shedding has been observed for the Murine Leukemia Virus (MLV) Env (*Riedel et al., 2017*; *Sjöberg et al., 2017*). And a recent study on the activation of pre-fusion HIV-1 gp41 to downstream intermediate states suggested that gp41 also opens through asymmetric intermediates (*Khasnis et al., 2016*). Our work illustrates the power of smFRET to define prevalent functional intermediates in the opening of the HIV-1 Env trimer.

## Materials and methods

### Cell lines

Name and Source: HEK293, ATCC, catalog # CRL- CRL-1573

Use and Authentication: The HEK293 cell line is used due to its high transfectability and gives high titers for produced retroviruses. The cell line is obtained as an early passage from ATCC which carries out Cell Authentication using Short Tandem Repeat Profiling before distribution. We are maintaining a big batch of frozen stocks from early passage. The cell line was tested and confirmed to mycoplasma free.

Name and Source: TZMBL, AIDS RRP # 8129

Use and Authentication: Used as indicator cell line for determining infectious units of produced HIV-1 viruses. We have tested the surface expression of CCR5 and CXCR4 using specific antibodies and challenging with R5 and X4-tropic viruses. We have also carried out PCR-based analyses to ascertain that the cell line is not infected with XMRV that was shown to increase infectivity of Nef-deficient viruses due to expression of glycoGag. The cell line was tested and confirmed to mycoplasma free.

### Preparation of labeled virions

The plasmid pNL4-3_V1Q3_V4A1 ΔRT encoding for HIV-1$_{NL4-3}$ Env and plasmid Q23_BG505_V1Q3-V4A1 ΔRT encoding for HIV-1$_{BG505}$ Env carrying the Q3 (GQQQLG) and A1 (GDSLDMLEWSLM) labeling peptides (*Wu et al., 2006*; *Yin et al., 2006*; *Zhou et al., 2007*) in V1 and V4 loops of full-length pNL4-3 ΔRT construct and full-length Q23_BG505 ΔRT construct, respectively; and the Env-expressing plasmid pCAGGS_JR-FL_V1Q3_V4A1 containing the peptides in analogous positions were previously described (*Munro et al., 2014*). The D368R point mutation (*Olshevsky et al., 1990*) was introduced into untagged constructs and V1Q3_V4A1 tagged constructs of full-length NL4-3, BG505 and Env-expressing JR-FL by overlap-extension PCR. All viruses were produced by co-transfecting HEK293 cells using the following ratios of plasmids:

HIV-1$_{NL4-3}$: 40:1 of pNL4-3 ΔRT to pNL4-3_V1Q3_V4A1 ΔRT;

HIV-1$_{BG505}$: 40:1 of Q23_BG505 ΔRT to Q23_BG505_V1Q3_V4A1 ΔRT;

HIV-1$_{JR-FL}$: 1:1 of the pNL4-3 ΔEnvΔRT and the Env-expressing constructs, which were at 40:1 for pCAGGS_JR-FL to pCAGGS_JR-FL_V1Q3_V4A1.

HIV-1$_{NL4-3}$_D368R: 40:1 of pNL4-3_D368R ΔRT to pNL4-3_V1Q3_V4A1_D368R ΔRT.

HIV-1$_{BG505}$_D368R: 40:1 of Q23_BG505_D368R ΔRT to Q23_BG505_V1Q3_V4A1_D368R ΔRT.

HIV-1$_{JR-FL}$_D368R: 1:1 of the pNL4-3 ΔEnvΔRT and the Env-expressing constructs, which were at 40:1 for pCAGGS_JR-FL_D368R to pCAGGS_JR-FL_V1Q3_V4A1_D368R.

HIV-1$_{NL4-3}$ mixed trimer 1: 40:1 of pNL4-3_D368R ΔRT to pNL4-3_V1Q3_V4A1 ΔRT.

HIV-1$_{BG505}$ mixed trimer 1: 40:1 of Q23_BG505_D368R ΔRT to Q23_BG505_V1Q3_V4A1 ΔRT.

HIV-1$_{JR-FL}$ mixed trimer 1: 1:1 of the pNL4-3 ΔEnvΔRT and the Env-expressing constructs, which were at 40:1 for pCAGGS_JR-FL_D368R to pCAGGS_JR-FL_V1Q3_V4A1.

HIV-1$_{NL4-3}$ mixed trimer 2: 20:20:1 of pNL4-3 ΔRT to pNL4-3_D368R ΔRT to pNL4-3_V1Q3_V4A1_D368R ΔRT.

HIV-1$_{BG505}$ mixed trimer 2: 20:20:1 of Q23_BG505 ΔRT to Q23_BG505_D368R ΔRT to Q23_BG505_V1Q3_V4A1_D368R ΔRT.

HIV-1$_{JR-FL}$ mixed trimer 2: 1:1 of the pNL4-3 ΔEnvΔRT and the Env-expressing constructs, which were at 20:20:1 for pCAGGS_JR-FL to pCAGGS_D368R to pCAGGS_JR-FL_V1Q3_V4A1_D368R.

Statistically, the 20:20:1 ratio for mixed trimer 2 will yield 50% trimers with the desired mixed trimer two shown in *Figure 2B*.

HIV-1$_{NL4-3}$ mixed trimer 3: 40:1 of pNL4-3 ΔRT to pNL4-3_V1Q3_V4A1_D368R ΔRT.

HIV-1$_{BG505}$ mixed trimer 3: 40:1 of Q23_BG505 ΔRT to Q23_BG505_V1Q3_V4A1_D368R ΔRT.

HIV-1$_{JR-FL}$ mixed trimer 3: 1:1 of the pNL4-3 ΔEnvΔRT and the Env-expressing constructs, which were at 40:1 for pCAGGS_JR-FL to pCAGGS_JR-FL_V1Q3_V4A1_D368R.

Virus was harvested 40 hr post transfection and concentrated by centrifugation over a 15% sucrose cushion at 20,000 x g for 2 hr. Virus pellets were resuspended in the labeling buffer containing 50 mM HEPES, 10 mM MgCl2, 10 mM CaCl2. Resuspended virus was enzymatically labeled with fluorophores in a reaction mixture containing Cy3B(3S)-cadaverine (0.5 μM), transglutaminase (0.65 μM; Sigma Aldrich), LD650-CoA (0.5 μM) (Lumidyne Technologies), and AcpS (5 μM) (*Zhou et al., 2007*). The labeling reaction was incubated at room temperature overnight. 0.1 mg/ml PEG2000-biotin (Avanti Polar Lipids) was added to the labeling reaction and incubated for 30 min at room temperature. Under these conditions, the labeling efficiencies of the Q3 and A1 tags are 40% and 55%, respectively (*Munro et al., 2014*). The virus was then purified away from excess fluorophore and lipid by ultracentrifugation (1 hr at 150,000 x g) over a 6–18% Optiprep (Sigma Aldrich) gradient containing 50 mM Tris pH 7.4 and 50 mM NaCl. The fractions containing viral particles were identified by p24 western blot and stored at −80℃.

## smFRET imaging

Labeled viruses were immobilized on a passivated quartz microscope slide coated with streptavidin. smFRET imaging was conducted at room temperature on a prism-based TIRF microscope with a 60 × 1.27 NA water-immersion objective (Nikon). Donor fluorophores were excited by a 532 nm laser (Laser Quantum) at ~0.1 kW per cm². Data were recorded at 25 frames per second for 2000 frames by two synchronized ORCA-Flash4.0v2 sCMOS cameras (Hamamatsu, 2048 × 2048), as described (*Juette et al., 2016*). The imaging buffer contained 50 mM Tris pH 7.4, 50 mM NaCl, and a cocktail of triplet-state quenchers. 2 mM protocatechuic acid (PCA) and 8 nM protocatechuic 3,4-dioxygenase (PCD) were also included to remove molecular oxygen (*Aitken et al., 2008*). For experiments performed in the presence of ligands, 0.1 mg/ml sCD4 or sCD4$_{D1D2}$-Igαtp were incubated with the immobilized virus at room temperature for 30 min prior to imaging.

## smFRET data analysis

The smFRET trajectories were extracted from the movies and processed using SPARTAN (*Juette et al., 2016*). FRET values were calculated according to FRET = $I_A/(\gamma I_D + I_A)$, where $I_A$ and $I_D$ are the fluorescence intensities of acceptor and donor dyes, respectively, and γ is the coefficient correcting for the difference in detection efficiencies of the donor and acceptor channels. The FRET histograms were fit to the sum of 3 Gaussian distributions. The occupancy in each FRET state was determined by the area under each Gaussian curve. Error bars were generated by propagating the uncertainties of the fits through the occupancy calculation. The occupancies in the FRET states were used to calculate the differences in free energies between states $i$ and $j$ according to $\Delta G°_{ij} = -k_B T ln(P_i/P_j)$, where $P_i$ and $P_j$ are the occupancies of the $i$th and $j$th state in the histogram, respectively, and $k_B$ is the Boltzmann constant.

smFRET traces were idealized using a segmental $k$-means algorithm with a 3-state model (**Qin, 2004**). The frequencies of transitions were displayed in transition density plots (TDP). Dwell times in each states were compiled into histograms and fit to the sum of two exponential distributions ($y = A_1 \exp^{-k1t} + A_2 \exp^{-k2t}$). The reported rates were determined by averaging the two rate constants, weighted by their respective amplitudes. The apparent change in activation energy barriers $\Delta\Delta G^{\ddagger}_{ij}$ was calculated according to $\Delta\Delta G^{\ddagger}_{ij} = -k_B T(k_{ij}^{liganded}/k_{ij}^{unliganded})$, where $k_{ij}$ is the rate constants determined from the exponential curves, $k_B$ is the Boltzmann constant, $T$ is the temperature in kelvin. The uncertainties of $\Delta G^{\circ}_{ij}$ and $\Delta\Delta G^{\ddagger}_{ij}$ were determined by propagating the uncertainties from the curve fitting.

## Infectivity measurements

Viruses carrying wild-type, D368R mutant HIV-1 Env were generated by transfection of HEK293 cells with full-length WT or mutant pNL4-3 or Q23_BG505 constructs, or at a 1:1 ratio of pNL4-3 ΔEnv and WT or mutant pCAAGS_JR-FL, along with 1:6 ratio of HIV-1-InGluc to viral constructs, using Fugene 6 (Promega, Madison, WI). Supernatants from 24 and 48 hr post-transfection were combined, filtered through a 0.45 μM filter (Pall Corporation) and titered on TZMbl cells. Titers were determined 48 hr post-infection either by measuring luciferase activity using the Gaussia luciferase assay kit (NEB). For testing the effects of ligands on infectivity, viruses were incubated with ligands for 30 min at room temperature prior to addition to TZMbl cells.

The V1V4-tagged BG505 was validated in form of the 100% tagged virus, not as the dye-labeled virus since the incomplete labeling efficiencies would leave enough unmodified trimers on the surface of a virus to account for nearly undiminished infectivity. While we cannot exclude additional effects of the dyes, we have not observed anisotropy for singly labeled virus, the dyes are highly hydrophilic (**Zheng et al., 2014**), do not associate with the viral membrane and no dye associated with viruses in the absence of labeling tags (**Munro et al., 2014**).

## Acknowledgements

We thank Alon Herschhorn and Joseph Sodroski for discussions. Patent applications pertaining to this work are U.S. Patent Application 13/202,351, Methods and Compositions for Altering Photophysical Properties of Fluorophores via Proximal Quenching (SCB, ZZ); U.S. Patent Application 14/373,402 Dye Compositions, Methods of Preparation, Conjugates Thereof, and Methods of Use (SCB, ZZ); and International and US Patent Application PCT/US13/42249 Reagents and Methods for Identifying Anti-HIV Compounds (SCB, JBM, WM). SCB is a co-founder of Lumidyne Corporation.

## Additional information

### Competing interests

Walther Mothes: Patent applications pertaining to this work are U.S. Patent Application 13/202,351, Methods and Compositions for Altering Photophysical Properties of Fluorophores via Proximal Quenching (S.C.B., Z.Z.); U.S. Patent Application 14/373,402 Dye Compositions, Methods of Preparation, Conjugates Thereof, and Methods of Use (S.C.B., Z.Z.); and International and US Patent Application PCT/US13/42249 Reagents and Methods for Identifying Anti-HIV Compounds (S.C.B., J. B.M., W.M.). S.C.B. is a co-founder of Lumidyne Corporation. The other authors declare that no competing interests exist.

### Funding

| Funder | Grant reference number | Author |
| --- | --- | --- |
| National Institutes of Health | GM116654 | Walther Mothes |
| National Institutes of Health | AI116262 | James B Munro |
| National Institutes of Health | GM098859 | Scott C Blanchard |
| National Institutes of Health | GM056550 | Scott C Blanchard Walther Mothes |

| Cancer Research Institute | Irvington Fellows Program | James B Munro |
| National Institutes of Health | AI042853 | James B Munro |
| China Scholarship Council | Yale World Scholars | Xiaochu Ma |
| National Institutes of Health | AI005023-17 | Peter D Kwong |

The funders had no role in study design, data collection and interpretation, or the decision to submit the work for publication.

### Author contributions

Xiaochu Ma, Conceptualization, Data curation, Formal analysis, Visualization, Writing—original draft, Writing—review and editing; Maolin Lu, Data curation, Formal analysis, Writing—review and editing; Jason Gorman, Conceptualization, Data curation, Methodology, Writing—review and editing; Daniel S Terry, Data curation, Software; Xinyu Hong, Software; Zhou Zhou, Hong Zhao, Roger B Altman, Methodology; James Arthos, Resources; Scott C Blanchard, Resources, Methodology, Writing—review and editing; Peter D Kwong, Conceptualization, Resources, Supervision, Funding acquisition, Writing—review and editing; James B Munro, Conceptualization, Formal analysis, Supervision, Funding acquisition, Methodology, Writing—original draft, Writing—review and editing; Walther Mothes, Conceptualization, Formal analysis, Supervision, Funding acquisition, Investigation, Writing—original draft, Project administration, Writing—review and editing

### Author ORCIDs

Scott C Blanchard http://orcid.org/0000-0003-2717-9365
Walther Mothes http://orcid.org/0000-0002-3367-7240

### Decision letter and Author response

Decision letter https://doi.org/10.7554/eLife.34271.021
Author response https://doi.org/10.7554/eLife.34271.022

## Additional files

**Supplementary files**

• Transparent reporting form
DOI: https://doi.org/10.7554/eLife.34271.019

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
