## [Decision Letter]

Thank you for submitting your article "HIV-1 Env trimer opens through an asymmetric intermediate in which individual protomers adopt distinct conformations" for consideration by *eLife*. Your article has been favorably evaluated by Wenhui Li (Senior Editor) and three reviewers, one of whom is a member of our Board of Reviewing Editors. The following individual involved in review of your submission has agreed to reveal his identity: Joseph D Puglisi (Reviewer #2).

The reviewers have discussed the reviews with one another and the Reviewing Editor has drafted this decision to help you prepare a revised submission.

The manuscript of Ma et al. addresses the longstanding debate about the conformational state of the HIV-1 Env, and the relationship of conformation to viral entry. Clearly, this is a conformationally flexible system involving challenging protein chemistry; as such, traditional structural approaches have led to conflicting models for the pathway of Env recognition by CD4 and co-receptors, and subsequent conformational steps along the viral entry pathway. Here the authors continue their elegant single-molecule FRET experiments applied to the Env protein on intact virions. They build on prior FRET measurements to assign the nature of the three distinct FRET states observed previously. They use clever choices of mutants, and ligand complexes to illuminate the conformational states of Env, and provide a low resolution cryo-EM structure that they claim supports their key observation – that Env opens through a key asymmetric intermediate involving changes in a single protomer.

The biophysical studies are an elegant mixture of single-molecule biophysics and challenging biochemistry/virology that deserves publication in *eLife* after changes that would greatly improve the presentation. However, our recommendation is that the EM structure should either be removed from this paper or extensively revised, which would require additional refinement and analyses of the structure.

We have the following suggestions for revising the smFRET part of the paper.

1) The authors use a challenging biochemical approach using orthogonal peptide tagging to allow double labeling of the individual protomers, combined with careful titration of unlabeled Env through co-expression. They have presented this approach previously, but it would have been useful for readers to know the extent of labeling efficiency for the virions. Also as described in the first paragraph of the subsection “State 3 corresponds to the gp120 conformation of the three-CD4-bound HIV-1 Env trimer”, the authors validate that insertion of the Q3 and A1 tags for fluorophore labeling did not affect Env functions using various assays, but it is not clear whether they tested labeled viruses in these assays, or viruses with just the peptide insertions. The peptide tag is just one component, and perhaps not the largest one, that distinguishes an unlabeled Env from a labeled Env. If it is not possible to do the validation assays using labeled viruses, then the authors should discuss caveats associated with possible steric occlusion from adding the dyes and whatever other components must be added to attach dyes to the peptide tags.

2) The use of a dodecameric CD4 oligomer (sCD4_D1D2_-Igαtp) for capturing state 3 in NL4-3 is interesting and promotes different effects than sCD4 in the 3 trimers analyzed by smFRET. It is unclear, however, if the "high local CD4 density" resulting from the architecture of this dodecameric CD4 oligomer would geometrically allow multiple copies of CD4 to engage a single Env? A cartoon showing the structure of the construct used in these experiments demonstrating this is feasible would address this and support the results presented. Figure 1D indicates that sCD4_D1D2_-Igαtp stabilizes state 3 for HIV-1(NL4-3), which corresponds to the open conformation with 3 CD4 bound, whereas the authors also claim that a single CD4 bound to the mixed trimer 1 leads to the same result (Figure 2D). How can these results be reconciled? And were smFRET studies done using sCD4_D1D2_-Igαtp on mixed trimer 1 (labeled protomer that can bind CD4 with two protomers that can't bind CD4)? This would allow a direct comparison of sCD4_D1D2_-Igαtp effects on native HIV-1NL4-3 trimer and mixed HIV-1NL4-3 trimer 1. As shown in Figure 2—figure supplements 3-5, the effects of sCD4_D1D2_-Igαtp were evaluated for JR-FL and BG505 viruses but not for NL4-3. Minor suggestion: it would be easier to compare if Figure 2—figure supplements 3-5 were combined into one figure.

3) The schematics used in Figure 1, 2 and 3 are confusing. It is hard to see the donor and acceptor, and it's not clear why they should have different FRET values in the different conformational states. It's also hard to see the difference between closed and open on such a small figure. Maybe using axial lines for the 3 protomers, and making the dyes bigger (and their positions different in the states) would help. These are the key figures for understanding all the single-molecule data, so the authors should spend more time on presentation. In particular, in Figure 1K, the donor and acceptor dyes are shown as being the same distance apart whether they are on the closed or the open trimer. This is confusing, since the dyes would be separated by long distances in the low FRET State 1 but by shorter distances in the intermediate and high FRET states 3 and 2. There is little discussion in this paper as to whether it makes structural sense for the dyes to become closer together upon CD4 binding, which is especially important if the authors want to include the cryo-EM structure of Env in the same paper. In particular, this sentence, "Binding of a single CD4 may loosen interaction of the V1/V2 loops in the trimer association domain so that neighboring protomers can adopt a conformation in which the V1 and V4 loops are closer to each other." Is this consistent with what is known about the relative locations of V1 and V4 in closed and open Env structures?

4) Figure 3I. Why is 17b shown as binding to the protomers in a mixed trimer that can't bind CD4? 17b won't bind to most Envs in the absence of CD4. It seems reasonable that 17b could bind to the protomer(s) that can bind CD4 in a mixed trimer, but what evidence is there that 17b can also bind to the CD4-incompetent protomers in a mixed trimer?

5) The authors should present prior measurements on CD4 affinity/kinetics wherever possible.

6) Figure 1 legend. Why are the authors using standard errors here instead of standard deviations as they used for other error analyses?

7) For FRET traces, it would be nice to show with lines the high, medium, and low states in each trace.

8) Materials and methods subsection “Preparation of labeled virions”. The protocol for use of the enzyme AcpS for labeling of one of the peptide tags should be cited.

9)Discussion, fifth paragraph. The authors say they have compared smFRET values for labeled BG505 virus and labeled BG505 SOSIP in a paper cited as Lu, Ma et al., 2017, under review. As the current submitted paper includes smFRET data for BG505 virus and an EM structure of a BG505 SOSIP, the paper under review is directly relevant to evaluating the submitted paper and should be given to the reviewers.

Cryo-EM structure

The structure itself (both results and methods) is problematic, and there is little effort to correlate the EM and smFRET results. We can't figure out why the authors think that the EM structure validates the smFRET results because the EM methods are inadequately described and/or possibly done incorrectly, the figures are poor quality, and the description of the structure is vague. We think this paper would stand alone as an important contribution without the EM structure, but if the authors want to include the EM, the following issues must be addressed before publication in any journal:

1) The Materials and methods section describing the EM procedures lacks a lot of information. How many particles were collected for the untilted and for the tilted data sets? How many were kept from each dataset after each step of classification? How many frames were collected in each movie? How many classes were used during 3D classification? How was CTF correction achieved for the two datasets; in particular, how was this done for the tilted data? Was dose-weighting used with motioncor2? What was the total exposure time and how many sub-frames was the total dosage divided into? The authors should use the ab initio model as a reference to do 3D classification, and detailed information about 3D classes should be included in the Methods and also shown in a supplementary figure. Additionally, since 2D/3D classification during cryoEM processing often leads to focusing on the "best" class(es), the authors should address whether there are other conformation(s) present in the dataset.

2) In Figure 4—figure supplement 1C. The authors only show 4 representative 2D classes (all of which look bad and appear to exhibit orientation bias, but this may be due to low resolution of the figure). Did they only use the particles belonging to these 4 classes as input for ab initio model generation? The authors should describe more details about the ab initio model generation input parameters for cryoSPARC in the Materials and methods section, they should use the ab initio model as a reference to do 3D classification, and detailed information about 3D classes should be included in the Materials and methods and also shown in a supplementary figure. Figure 4—figure supplement 4F should show the entirety of gp41 and gp120 rigid-body docked into the EM density, rather than just representative segments. Figure 4—figure supplement 1E is labeled as Map-to-model FSC but the legends states it is the gold standard FSC from refinement. It can only be one of the two.

3) What model was used as a reference for 3D refinement and was the reference model low pass filtered? The authors should verify that they are using the gold-standard FSC resolution estimation that was generated after post processing because this step would generate a more reliable resolution estimation. This information is especially important to know in the case of this structure because the map density and fitting in Figure 4—figure supplement 1F look worse than an 8Å resolution structure. Indeed, the resolution claim of 8Å seems optimistic since the gp41 inner helices are likely the best resolved part of the map and the one the authors chose to show. As a result, the reader cannot judge how confident the authors can be about the different CD4 orientation detected compared to the higher resolution 3 CD4-bound structure. This is further confounded by the strong preferred orientation of the sample in vitreous ice where each 2D class appears to be of the same orientation.

4) Materials and methods. We don't understand this sentence: "Atomic models of gp120 in the CD4-bound conformation, and 4-domain CD4, were fitted into the cryo-EM density by rigid body docking…" This implies that CD4-bound gp120 coordinates were fit into all three protomers in the cryo-EM map, but only one protomer is bound to CD4 in this structure. Also, the authors should discuss whether all four domains of CD4 were ordered in the density. It seems unlikely that D3 and D4 were as well-ordered as D1-D2 since CD4 exhibits some flexibility at the D2-D3 interface. Indeed, D2 of CD4 was not well-resolved in the cryo-EM structures of CD4-bound Env cited by the authors. How CD4 was fit is important because in Figure 4C, the authors used the fitted model to describe a CD4 D1-D2 domain orientation change. However, since full-length sCD4 is presumably flexible, this calculation would be unreliable if based on fitting all four domains. The authors should try lowering the contour level of the map and fitting only D1D2 or maybe only D1 and then do the comparison. Finally PDB codes should be cited for all coordinates used.

5) In order to publish this structure, the authors must show a figure with local resolutions plotted onto the structure. Without this, it is impossible to interpret sentences like this: "…the two adjacent ligand-free gp120 protomers are much more disordered than the CD4-bound protomer. At this resolution, the precise conformational state of the gp120 protomers could not be resolved." First, it's not clear what the resolutions of the ligand-free gp120 protomers actually are (need to see the requested figure to assess). Second and most importantly, if the precise conformational state of the ligand-free gp120 protomers cannot be resolved, then how is the cryo-EM structure relevant to interpreting the smFRET results?

6) The quality of Figure 4 is poor and it was difficult/impossible for all three reviewers to confirm the claimed asymmetry of the Env trimer. Additionally, it is also very difficult to see any of the proposed conformational changes on panels C (right) and D.

7) If the authors decide to retain the cryo-EM structure in this paper, they should provide the map mrc file as well as fitted coordinates file for the reviewers to better judge the cryo-EM map and the model building.

8) "The model of a gradual opening through asymmetric intermediates is supported by cryo-EM that depicts a single CD4 molecule bound to the DS-SOSIP.664 that cannot open, and the current structure with SOSIP.664 that can open in response to CD4…" We don't follow this logic. The existence of the previously published DS-SOSIP/single CD4 cryo-EM structure that is closed is not relevant to this model because the DS-SOSIP is prevented from opening due to introduction of a S-S bond, therefore what is the evidence that an Env without the added DS disulfide bond would remain closed?

9) Related to the point 8 above, Figure 4B shows a hypothetical progression of Env conformations from closed, to a single CD4-bound closed Env (the DS-SOSIP/single CD4 structure), to the current structure (described in the figure as single CD4-bound open structure), to the published 3 CD4-bound, open Env structure. This figure is misleading because it implies that the closed DS-SOSIP/single CD4 structure is a relevant intermediate (see above) and that all three protomers in the current structure are open. The current structure is said to "open in response to CD4" – is that what this EM structure shows? We can't tell from the figures. Figure 4B describes the current structure as a "single CD4-bound open structure" which might be interpreted as meaning that all three protomers adopt the open conformation. But in the Results section, the structure is referred to as an "asymmetric, open structure" which might mean that only the CD4-bound gp120 adopts the open conformation. "…the CD4 molecule is bound solely to one protomer in an open trimer configuration…" and later "…the current structure where the SOSIP.664 partially opens in response to binding of a single CD4…" These examples show that the text is confusing and possibly contradictory. Is the Env trimer in this structure "open" or is it "partially open"? Is the structure asymmetric or symmetric? The figures can't be used to address these questions, and since we don't know how disordered the non CD4-bound protomers are (point 9 above), the EM structure detracts from this paper.

---

## [Author Response]

The biophysical studies are an elegant mixture of single-molecule biophysics and challenging biochemistry/virology that deserves publication in eLife after changes that would greatly improve the presentation. However, our recommendation is that the EM structure should either be removed from this paper or extensively revised, which would require additional refinement and analyses of the structure.

We like to note up front that Peter Kwong and Priyamvada Acharya have agreed to remove the EM structure from the manuscript. Peter Kwong and Jason Gorman have supported our project on other aspects, and thus will remain authors. However, all authors involved in the structural work have agreed to be removed from the authors list, including Priyamvada Acharya.

We have the following suggestions for revising the smFRET part of the paper.1) The authors use a challenging biochemical approach using orthogonal peptide tagging to allow double labeling of the individual protomers, combined with careful titration of unlabeled Env through co-expression. They have presented this approach previously, but it would have been useful for readers to know the extent of labeling efficiency for the virions. Also as described in the first paragraph of the subsection “State 3 corresponds to the gp120 conformation of the three-CD4-bound HIV-1 Env trimer”, the authors validate that insertion of the Q3 and A1 tags for fluorophore labeling did not affect Env functions using various assays, but it is not clear whether they tested labeled viruses in these assays, or viruses with just the peptide insertions. The peptide tag is just one component, and perhaps not the largest one, that distinguishes an unlabeled Env from a labeled Env. If it is not possible to do the validation assays using labeled viruses, then the authors should discuss caveats associated with possible steric occlusion from adding the dyes and whatever other components must be added to attach dyes to the peptide tags.

Each new HIV-1 isolate that is being introduced for smFRET imaging was carefully validated as shown in this manuscript for the BG505. This is done with the 100% tagged Env.

With respect to labeling efficiencies, we have previously determined the labeling efficiencies and found them to be 40% for the Q3 tag and 55% for the A1 tag (Munro et al., 2014). We are still using the same protocols and have not observed a drop in the number of labeled particles in each preparation. Given that labeling efficiencies are not 100%, it’s not valid to test the infectivity of “100% labeled” virus since there are several trimers on the surface of the virus and incomplete labeling leaves enough unlabeled Env to maintain similar infectivity as WT. Thus, the test of Q3 and A1 peptide insertion on virus infectivity was done solely on viruses carrying 100% labeling peptides, but not on fluorophore-labeled viruses. We now mention this in the last paragraph of the Materials and methods subsection “Preparation of labeled virions”. While dyes could affect the outcome, which we acknowledge in the last paragraph of the Materials and methods subsection “Infectivity measurements”, we have not observed anisotropy for singly labeled virions suggesting no trapping of dyes in specific environments. These dyes are highly hydrophilic (Zheng et al., 2014). They also do not interact with the viral membrane. The association of dyes with virions is entirely dependent on the presence of labeling tag (Munro et al., 2014).

For us most important have been the biological controls. The smFRET signal has been responsive to ligands, mutations in a manner that correlates with virological data (Munro et al., 2014; this paper; Herschhorn et al., 2016). Moreover, the Tier 2 viruses JR-FL and BG505 are more closed and less responsive to sCD4 than the Tier 1 lab-adapted HIV-1 isolate NL4-3 (Figures 1B, 1E and 1H).

2) The use of a dodecameric CD4 oligomer (sCD4_D1D2_-Igαtp) for capturing state 3 in NL4-3 is interesting and promotes different effects than sCD4 in the 3 trimers analyzed by smFRET. It is unclear, however, if the "high local CD4 density" resulting from the architecture of this dodecameric CD4 oligomer would geometrically allow multiple copies of CD4 to engage a single Env? A cartoon showing the structure of the construct used in these experiments demonstrating this is feasible would address this and support the results presented. Figure 1D indicates that sCD4_D1D2_-Igαtp stabilizes state 3 for HIV-1(NL4-3), which corresponds to the open conformation with 3 CD4 bound, whereas the authors also claim that a single CD4 bound to the mixed trimer 1 leads to the same result (Figure 2D). How can these results be reconciled? And were smFRET studies done using sCD4_D1D2_-Igαtp on mixed trimer 1 (labeled protomer that can bind CD4 with two protomers that can't bind CD4)? This would allow a direct comparison of sCD4_D1D2_-Igαtp effects on native HIV-1NL4-3 trimer and mixed HIV-1NL4-3 trimer 1. As shown in Figure 2—figure supplements 3-5, the effects of sCD4_D1D2_-Igαtp were evaluated for JR-FL and BG505 viruses but not for NL4-3. Minor suggestion: it would be easier to compare if Figure 2—figure supplements 3-5 were combined into one figure.

The dodecameric CD4 oligomer was generated and extensively studied by the group of James Arthos (Bennet et al., 2007; Arthos et al., 2002). The interaction of the sCD4_D1D2_-Igαtp with native virions has been directly visualized by Sriram Subramaniam using cryo-electron tomography. 12xCD4 was found to cover several gp120 subunits of the same trimer as well as neighboring trimers highlighting the strong avidity (Bennet et al., 2007). The off-rate for sCD4_D1D2_-Igαtp from viruses is basically zero (Arthos et al., 2002).

In Figure 2D, however, only one protomer in a trimer is capable of binding to CD4. This therefore leads to a single-CD4-bound asymmetric trimer, regardless of whether the ligand is a single D1D2 CD4 molecule or sCD4_D1D2_-Igαtp. Viruses carrying the D368R mutation are also resistant to sCD4_D1D2_-Igαtp (Figure 2—figure supplement 3). Thus, upon CD4 binding, this single gp120 adopts the conformation of State 3 that corresponds to the CD4-bound conformation.

The reason why we use either sCD4 or sCD4_D1D2_-Igαtp is because the Tier 1 lab-adapted HIV-1 isolate is highly responsive and sensitive to sCD4, whereas the Tier 2 viruses JR-FL and BG505 are not. To trigger the CD4 bound conformation, we have to use the more potent ligand 12xCD4 for both Tier 2 isolates. If we use sCD4_D1D2_-Igαtp on NL4-3, it would be more than 1000x above the inhibitory concentration. We are trying to image all ligands at ~10x above the IC90. It wouldn’t be wise to test one ligand at ~10x above an IC90 and another at 1000x above the IC90.

Upon request, we have combined Figure 2—figure supplements 3-5 into one single Figure 2—figure supplement 3 for better comparison. And have added the figure of 100% D368R NL4-3 in Figure 2—figure supplement 3A.

3) The schematics used in Figure 1, 2 and 3 are confusing. It is hard to see the donor and acceptor, and it's not clear why they should have different FRET values in the different conformational states. It's also hard to see the difference between closed and open on such a small figure. Maybe using axial lines for the 3 protomers, and making the dyes bigger (and their positions different in the states) would help. These are the key figures for understanding all the single-molecule data, so the authors should spend more time on presentation. In particular, in Figure 1K, the donor and acceptor dyes are shown as being the same distance apart whether they are on the closed or the open trimer. This is confusing, since the dyes would be separated by long distances in the low FRET State 1 but by shorter distances in the intermediate and high FRET states 3 and 2. There is little discussion in this paper as to whether it makes structural sense for the dyes to become closer together upon CD4 binding, which is especially important if the authors want to include the cryo-EM structure of Env in the same paper. In particular, this sentence, "Binding of a single CD4 may loosen interaction of the V1/V2 loops in the trimer association domain so that neighboring protomers can adopt a conformation in which the V1 and V4 loops are closer to each other." Is this consistent with what is known about the relative locations of V1 and V4 in closed and open Env structures?

We have modified the figures to schematically illustrate the changes in the dye distance. We can, however, not relate the FRET data to current structural models as it is not trivial and requires the determination of the FRET values observed in Env protein complexes characterized structurally, which requires a separate study. For this reason, we have also removed the EM structure from this manuscript.

4) Figure 3I. Why is 17b shown as binding to the protomers in a mixed trimer that can't bind CD4? 17b won't bind to most Envs in the absence of CD4. It seems reasonable that 17b could bind to the protomer(s) that can bind CD4 in a mixed trimer, but what evidence is there that 17b can also bind to the CD4-incompetent protomers in a mixed trimer?

Because the single protomer that engages CD4 in these asymmetric trimers is already in State 3 and there is no difference between the CD4-bound and the CD4/17b bound gp120 conformation, neither in our smFRET assay nor structurally in gp120 or in the SOSIP trimer in Ozorowski et al., 2017). Thus, 17b likely binds in trans to the neighboring protomers. Mechanistically 17b likely binds by capturing a preexisting State 3 conformation that is more frequently sampled in State 2 as compared to the more closed State 1. However, this is rather speculative. It could in principal be tested experimentally in mixed trimers by combining the D368R mutation with mutations that prevent 17b binding, but introducing several mutations into Env can lead to non-linear effects and phenotypes are increasingly difficult to interpret. We have therefore decided to stay away from this speculation.

Additional evidence can be found in previous publications such as (Herschhorn et al., 2016) demonstrating that Env mutants residing in State 2 more readily engage coreceptors, leading toward downstream conformations.

We have previously shown in the 2014 Munro paper that the frequently opening Tier 1 lab-adapted NL4-3 can open in response to 17b alone, while the more closed Tier 2 JR-FL needs CD4.

5) The authors should present prior measurements on CD4 affinity/kinetics wherever possible.

We are working with native virions and measure infectivity, not affinities for recombinant proteins. We present the neutralization curves for CD4 and sCD4_D1D2_-Igαtp for all three HIV-1 isolates (Figures 2A, 2F and 2I; Figure 1—figure supplement 3). We now include a table with the calculated IC_50_ (Table 2), but we cannot include affinities.

6) Figure 1 legend. Why are the authors using standard errors here instead of standard deviations as they used for other error analyses?

Standard error= standard deviation/ (square root of sample size). We chose standard error for the estimation of data quality of histograms because it is more important for such a big number of data points that both reflect the mean and the accuracy of mean, which standard error takes into account. While for neutralization curves, there are only about 10 data points for each sample mean, therefore it is more important to present how each individual data point is different from the mean, which is reflected by standard deviation.

7) For FRET traces, it would be nice to show with lines the high, medium, and low states in each trace.

We have added lines indicating Low-, Intermediate- and High-FRET states in the FRET trace with Hidden Markov Modeling idealization (Figure 1A).

8) Materials and methods subsection “Preparation of labeled virions”. The protocol for use of the enzyme AcpS for labeling of one of the peptide tags should be cited.

We have added lines indicating Low-, Intermediate- and High-FRET states in the FRET trace with Hidden Markov Modeling idealization (Figure 1A).

9)Discussion, fifth paragraph. The authors say they have compared smFRET values for labeled BG505 virus and labeled BG505 SOSIP in a paper cited as Lu, Ma et al., 2017, under review. As the current submitted paper includes smFRET data for BG505 virus and an EM structure of a BG505 SOSIP, the paper under review is directly relevant to evaluating the submitted paper and should be given to the reviewers.

We have removed any reference to the Lu manuscript since the EM structure has been removed from this manuscript.

Cryo-EM structureThe structure itself (both results and methods) is problematic, and there is little effort to correlate the EM and smFRET results. We can't figure out why the authors think that the EM structure validates the smFRET results because the EM methods are inadequately described and/or possibly done incorrectly, the figures are poor quality, and the description of the structure is vague. We think this paper would stand alone as an important contribution without the EM structure, but if the authors want to include the EM, the following issues must be addressed before publication in any journal.

We have removed the EM structure from the manuscript.